# The value of genotype-specific reference for transcriptome analyses in barley

Wenbin Guo[1] , Max Coulter[2], Robbie Waugh[2,3] , Runxuan Zhang[1]

**It is increasingly apparent that although different genotypes within a species share "core" genes, they also contain variable numbers of "specific" genes and different structures of "core" genes that are only present in a subset of individuals. Using a common reference genome may thus lead to a loss of genotype-specific information in the assembled Reference Transcript Dataset (RTD) and the generation of erroneous, incomplete or misleading transcriptomics analysis results. In this study, we assembled genotype-specific RTD (sRTD) and common reference–based RTD (cRTD) from RNA-seq data of cultivated Barke and Morex barley, respectively. Our quantitative evaluation showed that the sRTD has a significantly higher diversity of transcripts and alternative splicing events, whereas the cRTD missed 40% of transcripts present in the sRTD and it only has ~70% accurate transcript assemblies. We found that the sRTD is more accurate for transcript quantification as well as differential expression analysis. However, gene-level quantification is less affected, which may be a reasonable compromise when a high-quality genotype-specific reference is not available.**

## Introduction

For more than a decade, RNA sequencing (RNA-seq) has become the preferred method for large scale transcript identification and quantification (Wang et al, 2009; Conesa et al, 2016). It accesses a more diverse collection of transcripts than earlier technologies like microarrays and allows studies of alternative splicing (AS) (Mantione et al, 2014; Conesa et al, 2016; Zhao, 2019). Consequently, one of the main uses of RNA-seq data is to quantify gene expression at both gene and transcript levels. Several studies have shown that for gene-level analysis, quantification at transcript resolution improves the overall estimation of gene expression (Trapnell et al, 2013; Zhao et al, 2015). Currently, the gold standard quantification of transcripts depends upon the use of well-annotated reference transcript datasets (RTDs) (Zhang et al, 2015, 2017; Brown et al, 2017; Rapazote-Flores et al, 2019) in conjunction with rapid and accurate computational programs that implement approaches based on *pseudo alignment*, such as Salmon (Patro et al, 2017) and Kallisto (Bray et al, 2016). De novo assembly methods can assemble transcripts without the guidance of a reference genome, but they suffer from significantly enhanced mis-assemblies and low sensitivities (Martin & Wang, 2011; Conesa et al, 2016; Marchant et al, 2016). High-quality transcript assembly is still frequently derived from reference genome mapping-based assembly approaches.

In the reference genome mapping-based approach, transcript assembly generally starts by mapping RNA-seq reads to a common reference; a haploid sequence considered representative of the genomes of related individuals within a phylogenetic clade (in our example this would be all domesticated barley genotypes). For example, the barley RTD BaRTv1.0 was constructed by mapping RNA-seq reads from more than 150 barley cultivars to the reference Morex genome (Hv_IBSC_PGSB_v2) (Rapazote-Flores et al, 2019). However, pan-genome studies have revealed that diverse genotypes contain a shared set of core genes as well as a large proportion (10–60%) of genotype-specific genes in subsets of individuals (Hirsch et al, 2014; Li et al, 2014; Golicz et al, 2016; Jin et al, 2016; Montenegro et al, 2017; Sun et al, 2017; Tao et al, 2019). Moreover, genomic sequences of common reference and individual strains contain frequent sequence variations, such as single nucleotide polymorphisms (SNPs), short deletions and insertions (INDEL), which also affect the transcript determinations in the assembly (Munger et al, 2014); sequence variation at splice sites will disrupt the recognition of introns and exons and alter protein translations (Anna & Monika, 2018; Baeza-Centurion et al, 2020).

Recently, high-quality genome assemblies for 20 genotypes of diverse geographical origin, spike morphology and annual growth habit have been made available through investigations into variation in the barley pan-genome (Jayakodi et al, 2020). In this study, ~1.5 million present/absent variants (PAVs) ranging from 50 to ~1M bp were identified and 5,602 deletions longer than 5 kb were found in Barke relative to Morex alone. Thus, the impact of moving from common reference to genotype-specific reference genome for transcriptomics studies, such as transcript assembly, quantification, and differential expression analysis could be profound. The availability of high-quality genotype-specific reference genomes

[1]Information and Computational Sciences, James Hutton Institute, Dundee, UK   [2]Plant Sciences Division, School of Life Sciences, University of Dundee at The James Hutton Institute, Dundee, UK   [3]Cell and Molecular Sciences, James Hutton Institute, Dundee, UK

Correspondence: runxuan.zhang@hutton.ac.uk

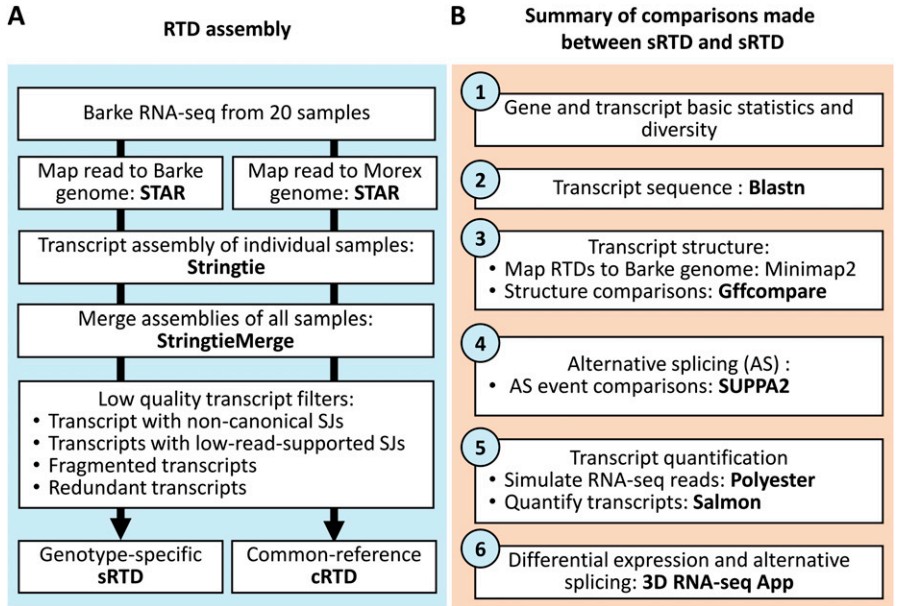

**Figure 1. Workflow.**
**(A)** RTD assembly pipeline with Barke RNA-seq read mapping to Barke and Morex genome, respectively. **(B)** Evaluation pipeline of the assembled sRTD and cRTD.

has created an opportunity to investigate the impacts of genotype-specific genetic variation on transcriptomics studies.

Here, we present a comprehensive investigation to quantitatively explore the benefits of using a genotype-specific reference genome for transcriptomics analysis in barley. We mapped RNA-seq data generated from Barke to the Morex genome (common reference; second release) (Mascher et al, 2017; Monat et al, 2019) as well as a newly assembled high-quality Barke genome (genotype-specific genome) (Jayakodi et al, 2020) and generated a common reference-based RTD (cRTD) and genotype-specific RTD (sRTD) in parallel using the same tools and parameters (Fig 1A). We evaluated the impact of using the sRTD in comparison with cRTD on transcriptome analysis using the following metrics: (1) gene and transcript diversity; (2) transcript sequences; (3) transcript structures; (4) AS; (5) quantification of transcript abundance; and (6) differential expression accuracy (Fig 1B).

## Results

### Evaluation of genome quality and the bias caused by the transcript assemblers

The barley pan-genome project showed that the Barke genome and Morex genome used in our study are of very similar quality with the Morex genome being slightly better in most metrics. They both had a high co-linearity and comparable presentation of gene models (Jayakodi et al, 2020). To further investigate the genome quality, we calculated the scaffold statistics of seven chromosomes and the unassigned contig in the two haploid reference genomes. The total scaffold sizes of Barke and Morex genomes are 4.2 billion and 4.3 billion. The N50 numbers are 605.6 million and 624.2 million, respectively. The unknown nucleotide bases in the Morex genome are slightly higher, with 2.72% compared with 0.96% in the Barke genome. The percentages of adenine (A), cytosine (C), guanine (G), and

thymine (T) are very close in the two reference genomes (Table S1). We also used BUSCO to evaluate the completeness of the genomes by using embryophyta_odb10 as the lineage dataset (Manni et al, 2021). The completeness of the two reference genomes is equally good. The complete (C) and single-copy (S) proportions of the Barke and Morex genome are 98.2% and 98.5%, respectively (Fig S1). Therefore, the discrepancy between sRTD and cRTD caused by reference genome quality is likely to be minimum.

In addition, we also investigate the transcript assemblies by using different assemblers. We used Cufflinks (Trapnell et al, 2010), Scallop (Shao & Kingsford, 2017), and Stringtie (Pertea et al, 2015) to perform three parallel transcript assemblies for 20 diverse tissue samples from the barley *cv.* Barke as described in Coulter et al (2021) *Preprint*. The transcripts using each assembler were merged by Stringtie-merge. The read mapping and quality control steps were the same as in Fig 1A. We identified problematic transcripts from a few categories, including transcripts with false splice junctions (SJs were not present at read alignments) and low read support SJs, redundant transcripts with an identical structure to a longer transcript in the same gene, transcripts likely to be fragmented (length <70% of the gene length) and unstranded transcripts (Table S2). We found that Cufflinks assembled significantly more problematic transcripts compared with Scallop and Stringtie. Cufflinks could generate up to 80% misassembled transcripts, including ~55% transcripts with false SJs. In addition, both Cufflinks and Scallop produced a high number of mono-exon antisense transcripts (mono-exon transcripts that overlap with a gene are in antisense), which is likely to be a result of mis-assembly from reads with incorrect strand information (Mourão et al, 2019). Also, Scallop generates 20,000 (50%) fewer transcripts compared with Stringtie. As the focus of this study is to investigate the impact of reference genome on transcriptomics analysis, to minimise the biases introduced by assemblers, we have focused on the Stringtie results in this study given this is the assembler that presented the best accuracy and sensitivity.

## Genome mapping comparisons

We used STAR to map Barke RNA-seq reads to the Barke and Morex genomes by allowing 0 mismatches (Dobin et al, 2013). The average mapping statistics of the samples are shown in Table S3. We can see that >8.6 million extra pairs of reads (7.26% more of the total) are uniquely mapped to the Barke genome compared with Morex. The percentage of unmapped reads is higher when mapping to the common reference (Morex). Most of the unmapped reads (>98%) are due to the mapping length being less than 2/3 of the mapped read length. STAR statistics show that 14.13% of total reads are unmapped in the common reference for this reason, whereas this number dropped to 6.90% in the genotype-specific (Barke) genome. The genotype-specific reference genome has fewer missing segments than the common reference and thus has a low percentage of unmapped reads. The spliced reads used to locate introns in the transcript assembly are also more abundant by >9.4 million pairs of reads (10.96%) in the genotype-specific alignment. Sequence variations including deletions and insertions, increase from 0.01 to 0.02% and 0.00 to 0.02%, respectively, with their average lengths longer when mapping to the common reference (Table S3).

To investigate the effect of increasing the tolerance to sequence variations on the read mapping and transcript assembly, we explored the allowances of 0, 2, 4 and 6 mismatches in the second pass of read mapping during sRTD and cRTD assembly (Tables S4–S6). We observed a general trade-off in that by allowing more mismatches, more reads are mapped but the chances of mismapping also increase. When mapping Barke RNA-seq reads to the Morex genome allowing two mismatches, 107,619,513 (89.78%) reads map uniquely, compared with 107,770,451.15 (89.89%) when mapping the same data to the Barke genome allowing 0 mismatches. We, therefore, used Gffcompare to assess differences in the transcript structures in a cRTD constructed allowing two mismatches, denoted as cRTD_m2 (Table S7). When allowing two mismatches, the number of missed gene loci in cRTD_m2 reduced by 1,979 from 21.5 to 18.2%. However, the number of false-positive loci increased by 3,600 from 13.6 to 18.6%. Thus, allowing two mismatches enables more genes to be discovered but this is accompanied by more false discoveries. For individual transcripts, we observed decreased precision at almost all levels when allowing mismatches, whereas recall increases at almost all levels. Thus, to focus on the impact of the reference genome and to minimise the confounding of mismatch settings, we used a cRTD constructed with the parameter setting of 0 mismatches for all the quantitative analyses. The errors and false discovery rates we use are therefore conservative estimates.

## Comparison of high-level statistics on sRTD and cRTD

We compared high-level statistics of various features, such as genome coverage, genes, transcripts, exons, and introns between sRTD and cRTD (Table 1). At the genome level, cRTD covers ~91.16 million bases on the genome, whereas sRTD covers 103.79 million bases, a 13.85% increase. This is consistent with the higher number of unmapped reads identified during the STAR mapping step (Table S3). At the gene level, sRTD and cRTD include a similar number of genes but sRTD has 8.69% more multi-isoform genes. Protein-

coding genes with significant hits in the UniProt Plant database (Bateman et al, 2021) are also 2.26% higher in sRTD (21,365 in sRTD versus 20,893 in cRTD). Differences at the transcript level are more profound. Despite having 0.36% fewer genes, the number of transcripts in sRTD increased to 144,872 compared with 128,438 in cRTD, a 12.8% (16,434) increase. The numbers of protein-coding transcripts and those with significant hits in the UniProt Plant database in the sRTD are 15.09% and 13.03% higher, respectively, than in the cRTD and the average protein-coding length is 16.43 (4.28%) longer. Transcript diversity in the sRTD also shows a 13.19% increase, rising to 2.44 from 2.15 transcripts per gene in cRTD (Table 1 and Fig S2). The sRTD tends to have more and longer exons with the average transcript length in sRTD 12.11% (268.56 bp) longer than in cRTD. With the effects of longer transcripts and increased transcript diversity, there is a 26.45% increase of the total transcript length (exonic) in sRTD over cRTD. With significantly less genome coverage, shorter transcripts and fewer protein sequences, it is likely that there are more transcript fragments and incomplete gene models in cRTD than in sRTD and the fragmentation and incompleteness of cRTD transcripts affect their coding capacity. Associated with the increased transcript diversity in sRTD we observe an increased number of AS events in all event categories. The increase ranges from 21.78% for alternative acceptor sites (A3) to 30.98% for mutually exclusive exons (MX) (Table S8).

## Transcript sequence and structure comparisons

We used Blastn (https://blast.ncbi.nlm.nih.gov/Blast.cgi) to compare the transcript sequences between sRTD and cRTD. To estimate the technical variations caused by Blastn itself, we also compared transcript sequences in sRTD to itself. We defined precision, recall and their weighted mean F1 score to evaluate the sequence similarity (Fig 2A). Blastn locally aligns stretches of sequences with a high level of base matches without considering diverging regions (Altschul et al, 1990). Thus, two transcript sequences may have significant Blastn e-value and high bit-score but have only a small proportion of overlap. Thus, we defined an "F1 score" to identify transcript pairs with significant sequence overlap proportional to the full length of sequences. At the nucleotide base level, on average 95.2% (precision) of the transcript nucleotides in cRTD are the same as sRTD. Less than 5% of transcript sequences are unique to cRTD. However, 11% of the nucleotide bases from sRTD are missing from the cRTD, indicating a loss of >10% of transcript sequence information when mapping RNA-seq reads to the common reference, consistent with the lower genome coverage identified above. At the individual transcript level, we used an F1 value of 0.78 (the lower limit of outliers of cRTD and sRTD sequence similarity, Figs 2A and S3A) as a cut-off to determine transcripts with a high proportion of sequence overlap. With this threshold, Blastn identified 98.8% of transcripts between sRTD and itself, indicating a low level of technical false positives and negatives (both at 1.2%) caused by Blastn and the chosen parameters. In comparison between cRTD and sRTD, we found that 39,920 (31.1%) transcripts in cRTD do not have a matched transcript of sufficiently high sequence similarity in sRTD. These are likely to be misassembled transcripts. 56,354 (38.9%) of the transcripts in the sRTD do not have a corresponding transcript in the cRTD, representing transcripts that failed to

**Table 1.** Basic statistics of the assembled sRTD and cRTD.

| Category | sRTD | cRTD | (sRTD–cRTD)/cRTD |
|---|---|---|---|
| Genome-covered bases | 103,793,515 | 91,164,519 | 13.85% |
| Gene number | 59,447 | 59,664 | −0.36% |
| Multi-isoform gene number | 16,163 | 14,871 | 8.69% |
| Mono-exon gene number | 32,833 | 33,001 | −0.51% |
| Multi-exon gene number | 26,614 | 26,663 | −0.18% |
| Protein-coding[a] | 33,254 (55.94%) | 31,961 (53.57%) | 4.05% |
| Best-hit in UniProt Plant (e < 0.01)[b] | 21,365 (35.94%) | 20,893 (35.02%) | 2.26% |
| Transcript number | 144,872 | 128,438 | 12.80% |
| Mono-exon transcript number | 33,429 | 33,578 | −0.44% |
| Multi-exon transcript number | 111,443 | 94,860 | 17.48% |
| Protein coding | 73,998 (51.08%) | 64,294 (50.06%) | 15.09% |
| Protein average length | 399.93 | 383.5 | 4.28% |
| Best-hit in UniProt Plant (e < 0.01) | 47,614 (32.87%) | 42,126 (32.80%) | 13.03% |
| Transcript number per gene | 2.44 | 2.15 | 13.19% |
| Transcript N50 | 3,284 | 3,056 | 7.46% |
| Transcript N90 | 1,457 | 1,286 | 13.30% |
| Transcript average length (exonic) | 2,486.78 | 2,218.22 | 12.11% |
| Transcript total length (exonic) | 360,264,709 | 284,903,601 | 26.45% |
| Exon number | 881,447 | 722,816 | 21.95% |
| Exon number per transcript | 6.08 | 5.63 | 8.10% |
| Exon average length | 408.72 | 394.16 | 3.69% |
| Intron number | 736,575 | 594,378 | 23.92% |
| Intron number per transcript | 5.08 | 4.63 | 9.85% |
| Intron average length | 643.8697 | 655.8377 | −1.82% |

[a]Protein-coding genes are defined as those genes with at least one protein-coding transcript.
[b]The number of genes that have at least one protein-coding transcript with the best-hit in the UniProt Plant database.

assemble when mapping to the common reference (Fig 2B). The transcript sequence discrepancy between cRTD and sRTD directly affects the translated proteins. Only 46,019 of 73,081 protein-coding transcripts in sRTD have a match in cRTD, indicating that 37% of the protein-coding transcripts are not assembled or with sufficient similarity to those in sRTD (Figs 2B and S3B). When summarised to the gene level, 20,007 (33.7%) genes in sRTD do not find any genes of sufficiently high transcript sequence similarity in cRTD, whereas 20,224 (33.9%) genes in cRTD have no corresponding gene models in sRTD. Only 72.3% of the protein-coding genes in sRTD have been retrieved in cRTD with at least one matched protein sequence (Fig 2C).

To analyse the transcript structure on the same genome coordinate system, we mapped the whole transcript sequences of sRTD and cRTD to the Barke genome with Minimap2 (Li, 2018), denoting as sRTD.minimap2 and cRTD.minimap2, respectively. Like Blastn analysis, the former was used as technical control to estimate the errors introduced by Minimap2. We used Gffcompare to evaluate the structure match of sRTD.minimap2 and cRTD.minimap2 to sRTD at various levels, including nucleotide bases, exon, intron, intron chain, transcripts and gene loci (Pertea & Pertea, 2020). In the

Gffcompare analysis, the overlapped gene models shared the same loci. Duplicate transcripts or transcripts whose segments were aligned to multiple chromosomes and/or strands by Minimap2 were discarded, which filtered out 247 transcripts in sRTD.minimap2 and 1,906 transcripts in cRTD.minimap2 (Table 2). The evaluation statistics of true positive was defined as those features that had identical boundaries between the RTDs for comparison (Fig S4). Minimap2 can accurately map the Barke-based assembly back to the Barke genome with minor technical errors of both false positives and false negatives at <2% at all levels (Tables 2 and 3). However, by using the common reference (Morex) for transcript assembly, we lose ~10% of introns and exons defined by their genomic coordinates and 20% of gene loci. We also have up to 13% of novel predictions that do not exist in the sRTD (Table 2). These observations indicate that the common reference genome poses a significant challenge to accurately determine the genomic locations of genotype-specific genes and transcripts, including the AS sites, alternative transcriptional starts as well as polyadenylation site selections.

At the transcript level, Gffcompare classified multi-exon transcripts with identical intron chain or mono-exon transcripts with

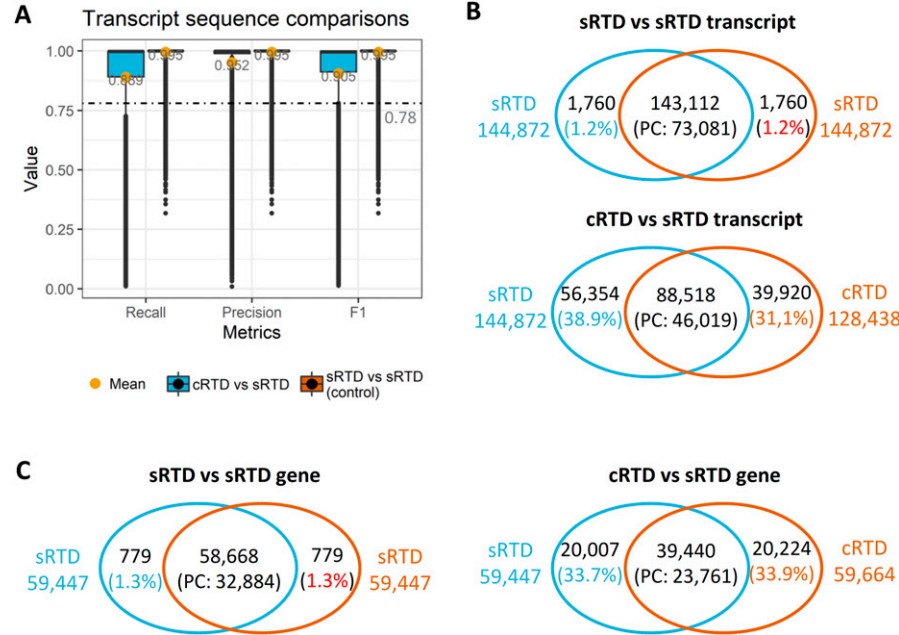

**Figure 2. Summary of transcript sequence comparisons of sRTD and cRTD.**
**(A)** Metrics distribution of transcript sequence comparisons. The transcript sequences of cRTD and sRTD (query) were both compared with sRTD (target) with Blastn. The recall was defined as the number of sequence matched bases over the target sequence length, whereas precision was the matched bases divided by query sequence length. F1 score was the harmonic mean of recall and precision. **(B)** Venn diagram of the transcript-to-transcript match. The overlaps of sRTD versus sRTD and cRTD versus sRTD were those transcripts with sequence similarity F1 > 0.78 (also see Fig S3). PC: protein-coding. **(C)** Gene level comparisons. The overlap represents the genes of matched transcript in (B).

**Table 2. Missed and novel exon, intron and loci regions of Barke genome.**

| Category | sRTD.minimap2[a] versus sRTD[b] | cRTD.minimap2[c] versus sRTD[b] |
|---|---|---|
| False negative | | |
| Missed exons | 1,158/291,963 (0.4%) | 31,391/291,963 (10.8%) |
| Missed introns | 304/172,487 (0.2%) | 18,213/172,487 (10.6%) |
| Missed loci | 805/59,440 (1.4%) | 12,787/59,440 (21.5%) |
| False positive | | |
| Novel exons | 448/291,743 (0.2%) | 12,815/259,285 (4.9%) |
| Novel introns | 132/172,398 (0.1%) | 4,360/154,670 (2.8%) |
| Novel loci | 354/58,990 (0.6%) | 7,848/57,517 (13.6%) |

[a]Query sRTD.minimap2: 144,625 in 58,990 loci (111,207 multi-exon transcripts).
[b]Reference sRTD: 144,872 transcripts in 59,440 loci (111,443 multi-exon).
[c]Query cRTD.minimap2: 126,532 transcripts in 57,517 loci (93,268 multi-exon transcripts).

**Table 3. Recall and precision of structure comparisons at transcript and transcript unit level.**

| Category | sRTD.minimap2 versus sRTD | | cRTD.minimap2 versus sRTD | |
|---|---|---|---|---|
| Statistics | Recall | Precision | Recall | Precision |
| Base level | 99.5% | 99.8% | 76.9% | 90.2% |
| Exon level | 99.3% | 99.4% | 74.1% | 81.4% |
| Intron level | 99.6% | 99.6% | 84.6% | 94.4% |
| Intron chain level | 98.8% | 99.0% | 57.6% | 68.6% |
| Transcript level | 98.6% | 98.6% | 56.8% | 64.6% |
| Locus level | 98.1% | 98.1% | 62.5% | 64.6% |

significant overlap (more than 80% to the longer transcript; default by Gffcompare) as equivalent transcripts. The matched genes must have at least one matched transcript (Fig S4). In Table 3, we can see that cRTD.minimap2 loses about 40% of the total transcripts, multi-isoform transcripts (intron chains) and gene loci present in sRTD and has about 35% novel assemblies where the structure cannot be found in sRTD, agreeing with the transcript sequence comparisons in Fig 2B. The detailed sub-categories of the transcript structure differences are shown in Fig S4. The largest categories of un-matched cRTD transcripts are fragments with (1) only one side of a splice junction correctly matched (8.70%); (2) correct splice junctions (7.63%); and (3) retained introns (7.18%). The metrics improve when scaled down to smaller units (nucleotide bases, exons and introns). The introns are the best retrieved, with 85% recall and 94%

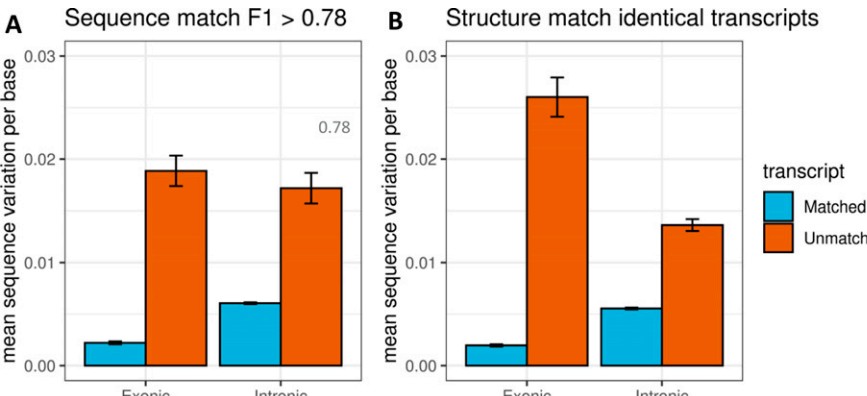

Figure 3. **Exonic and intronic sequence variations of the transcripts in the comparisons.**
For each transcript, the exonic and intronic sequence variation per base values were calculated with the total bases of variations (SNPs + insertions + deletions) in exons and introns divided by the transcript length, respectively. **(A)** The matched transcripts are determined by sequence similarity F1 score > 0.78. **(B)** The matched transcripts are those with identical structures determined in the structure comparison.

precision. The matched bases have slightly lower recall (77%) and precision (90%). Because of the difficulties in matching the boundaries of first and last exons, the statistics for the exons are the lowest, at 74% and 81%, respectively. The drastic decrease in recall and precision when combining the introns and exons into transcripts is mainly due to the complexity of AS and variations at the 5′ and 3′ sites.

By combining the sequence and structure comparisons, we identified 78,540 (61.2%) transcripts in cRTD that find an equivalent transcript in sRTD with a significant sequence as well as structure similarity (Table S9). We also observed disagreements between sequence and structure matches. We identified 1,776 transcripts that had identical structures in sRTD and cRTD, but with significantly different sizes of first and/or last exons, thereby leading to low sequence similarity of F1 ≤ 0.78 (Table S9 and Fig S5A). On the other hand, 9,978 transcripts with high sequence similarity (F1 > 0.78) had distinct structures, which were due to inconsistent intron sizes, such as the shift of intron boundaries, missing and novel introns caused by technical errors in Minimap2, or sequence variation arising from mapping Morex-based assemblies to the Barke genome (Table S9 and Fig S5B–D). Thus, the statistics for sequence and structure comparisons show a different level of similarities, especially for the nucleotide base level (Fig 2 and Table 3).

### Exonic and intronic sequence variations contribute to transcript discrepancy

To further explain how the underlying exonic and intronic sequence variations between Barke and Morex genomes contribute to the common and different transcripts with sRTD and cRTD, we extracted the gene loci sequences (including exonic and intronic sequences) of cRTD from the Morex genome and aligned these sequences to the Barke genome by using Minimap2. SNPs were identified with BCFtools (Danecek et al, 2021), whereas insertions and deletions (INDELs) were directly extracted from the cigar strings of the sequence alignment bam file. We calculated the exonic and intronic sequence variation (SNPs + insertions + deletions) per base of the common and different transcript sets in the sequence and structure comparisons. We found that common transcripts matched by high sequence similarity or identical structure between sRTD and

cRTD tend to have significantly lower sequence variations in exons and introns by comparing the Barke and Morex genomes. The common transcripts also have substantially higher sequence variations in the introns than in the exons. The unmatched transcripts between cRTD and sRTD have sequence variations that are 10 times greater in the exonic regions and three times greater in the intron regions (Fig 3A and B).

### AS event comparisons

By using the Minimap2 results, we can directly compare the AS events between cRTD.minimap2 and sRTD.minimap2 on their genomic coordinates. We used SUPPA2 to generate local AS events of retained intron (RI), alternative 5′ splice site (A5), alternative 3′ splice site (A3), skipping exon (SE), alternative first exon (AF), alternative last exon (AL), and mutually exclusive exons (MX) at the splice junctions of multi-exon transcripts (Trincado et al, 2018). The AS events of cRTD.minimap2 and sRTD.minimap2 are treated as queries and compared with the target events in sRTD (Fig 4A). The comparisons of sRTD.minimap2 and sRTD revealed minor technical errors for all the events, with Precision and Recall both close to 1. The precision and recall of cRTD.minimap2 against sRTD vary at different events. The A3, A5, and SE events are the best matched, but still, about 10% of splice junctions failed to be identified, and we observed a 30% false discovery rate. Most of the AF and AL events in cRTD.minimap2 are incorrect predictions, which is consistent with our earlier analysis that cRTD tends to contain short fragmented transcripts, which are likely to be mis-annotated at the transcript ends.

We further studied the percentage spliced-in (PSI) of sRTD.minimap2, cRTD.minimap2, and sRTD, a measure of the relative abundance of AS events. We used Salmon and the Barke RNA-seq reads of the 20 samples from Coulter et al (2021) *Preprint* to generate TPMs for event PSI calculation in SUPPA2. We found that the Pearson and Spearman correlations between PSI values in cRTD.minimap2 and identical events in sRTD were significantly lower than that in the control (comparison of sRTD.minimap2 and sRTD) for all the AS events, whereas the mean relative errors (absolute value of PSI difference divided by PSI of sRTD) of the 20 samples were significantly higher (Fig 4B–D). Thus, transcript assembly from the cRTD is particularly problematic when used for AS analysis.

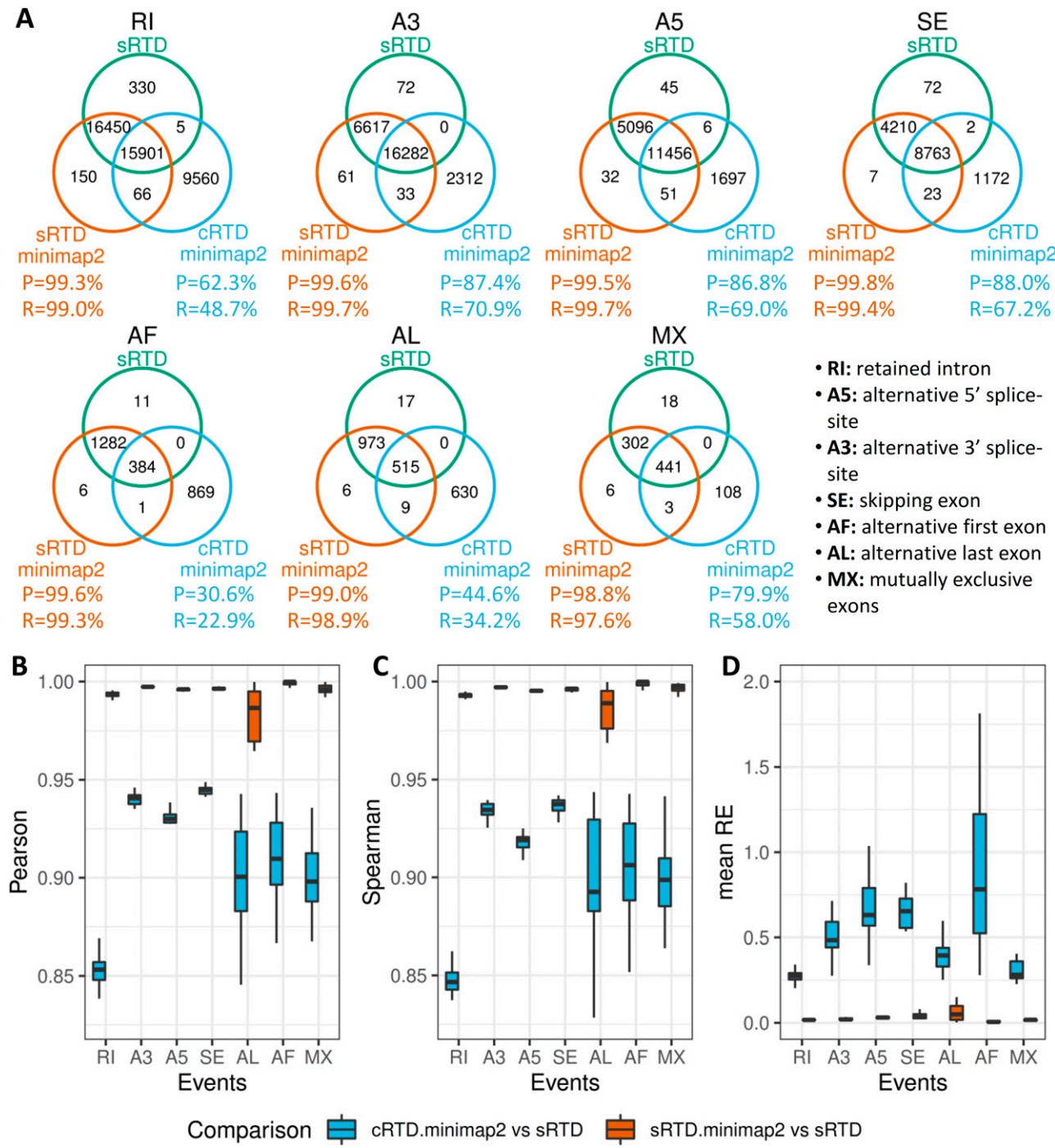

**Figure 4. Comparisons of alternative splicing (AS) analysis accuracy.**
**(A)** Comparisons of AS event numbers in sRTD.minimap2 and cRTD.minimap2 (query) against sRTD (target). Recall (R) = (target & query)/target, Precision (P) = (target & query)/query. **(B, C, D)** The Pearson correlation (B), Spearman correlations (C) and mean relative errors (D) were calculated for different AS events in 20 samples. The outliers of the distributions were removed.

## Transcript quantification comparisons

To investigate how the expression estimation for individual genes is affected and what are the characteristics of the genes that are most affected, we generated simulated RNA-seq reads to assess the

quantification errors introduced by using different RTDs. We used the read count matrix generated from Barke caryopsis and root tissues (each with three biological reps) from Jayakodi et al (2020) to simulate the RNA-seq data using the Polyester R package based on sRTD (Frazee et al, 2015). The corresponding transcripts between

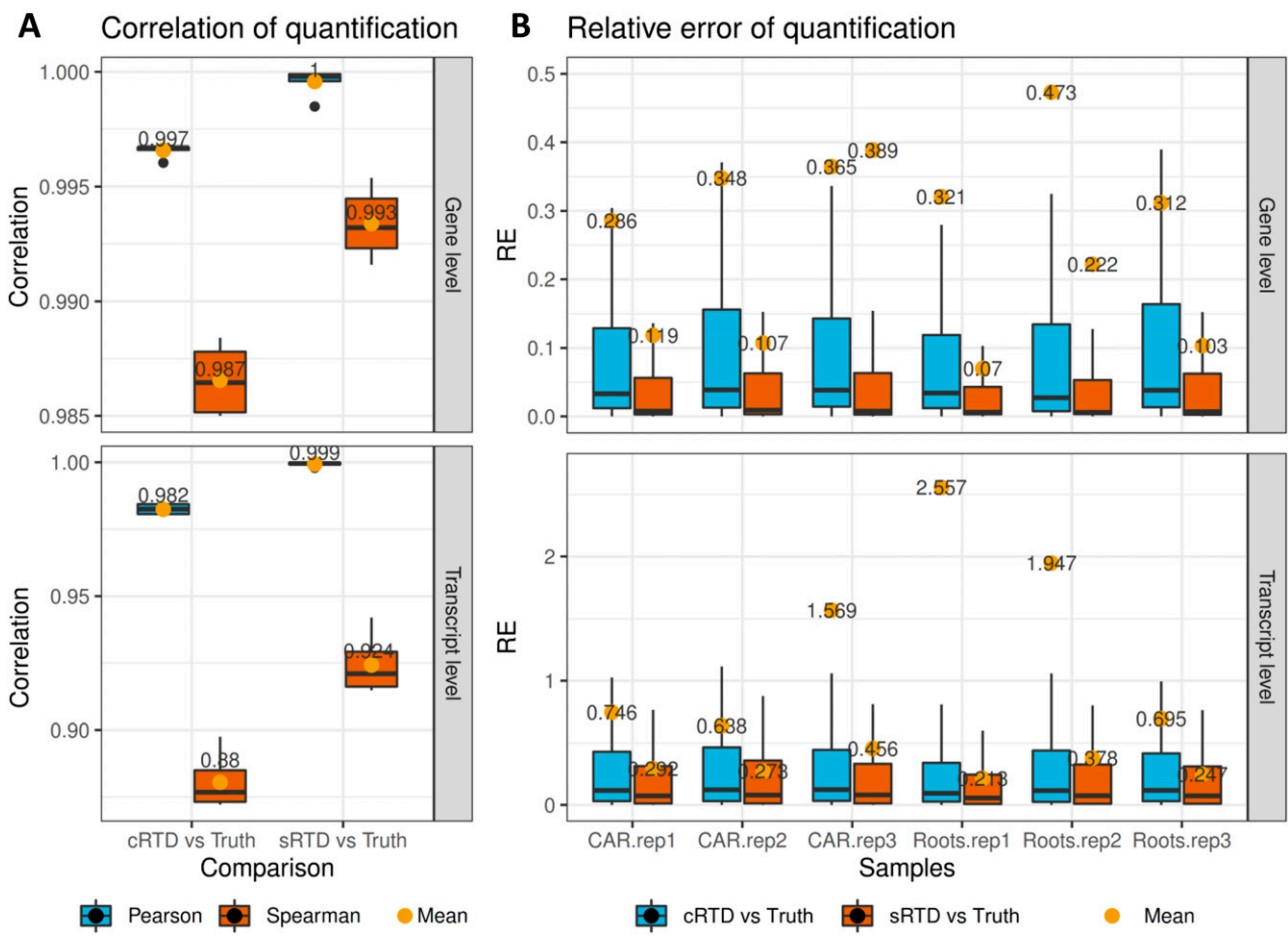

**Figure 5. Statistics of transcript and gene quantification comparisons.**
**(A, B)** The read counts at gene and transcript levels from sRTD and cRTD quantifications were both compared to ground truth for calculations of (A) Pearson and Spearman correlation of six samples and (B) relative errors (RE) in each sample. **(B)** The dots highlight the mean values of the distributions and the outliers in (B) are removed for better visualisation.

cRTD and sRTD were obtained using the Blastn sequence match with F1 score > 0.78 and Gffcompare match with identical transcript structures (Table S9). Thus, the transcript quantifications of cRTD and sRTD can be directly compared with the read count matrix (ground truth) used for the simulation. Although we used stringent criteria to define the equivalent transcripts, we find that the Pearson and Spearman correlations of comparisons between sRTD and ground truth are constantly higher than that of cRTD and ground truth (Fig 5A). The relative errors are significantly higher between cRTD and ground truth at both gene and transcript levels (Fig 5B). sRTD outperforms cRTD in all categories at both transcript and gene levels. The relative errors are two to threefold higher when using cRTD for quantification even at the gene level, indicating that sRTD provides more accurate quantification at both transcript and gene levels. The results also show that quantification is more accurate at the gene level than at the transcript level for the RNA-seq data.

The cRTD transcripts with lower similarities to sRTD (F1 score < 0.78) were also investigated in terms of quantification accuracy. The transcripts in cRTD were classified into eight groups according to their F1 scores in intervals of 0.1 between 0 and 0.78 (Fig 6). The relative error of the quantification increases from 0.5 (50%) to 1

(100%) as the sequence similarities between the corresponding transcript in cRTD and sRTD decrease (Fig 6A). The average correlation between the transcript abundance estimation using cRTD and the ground truth also improves (Pearson correlation from 0.367 to 0.81, Spearman correlation from 0.028 to 0.65) with an increasing similarity score (Fig 6B). Thus, transcripts in cRTD with sequences different from those in sRTD (38.9% of transcripts with F1 score <0.78) would produce significant quantification errors and should therefore be a cause of concern when used for quantification and differential expression analysis.

### Differential expression and AS

We further investigated the downstream impact on differential expression analysis with the simulated RNA-seq data. We used the 3D RNA-seq pipeline to investigate (1) differentially expressed (DE) genes and transcripts; (2) differentially alternatively spliced (DAS) genes; and (3) differential transcript usage (DTU) transcripts between the caryopsis and roots tissues (Guo et al, 2020). The PCA plots (Fig S6A–C) reveal that the quantifications obtained using both sRTD and cRTD successfully capture the variation in the data

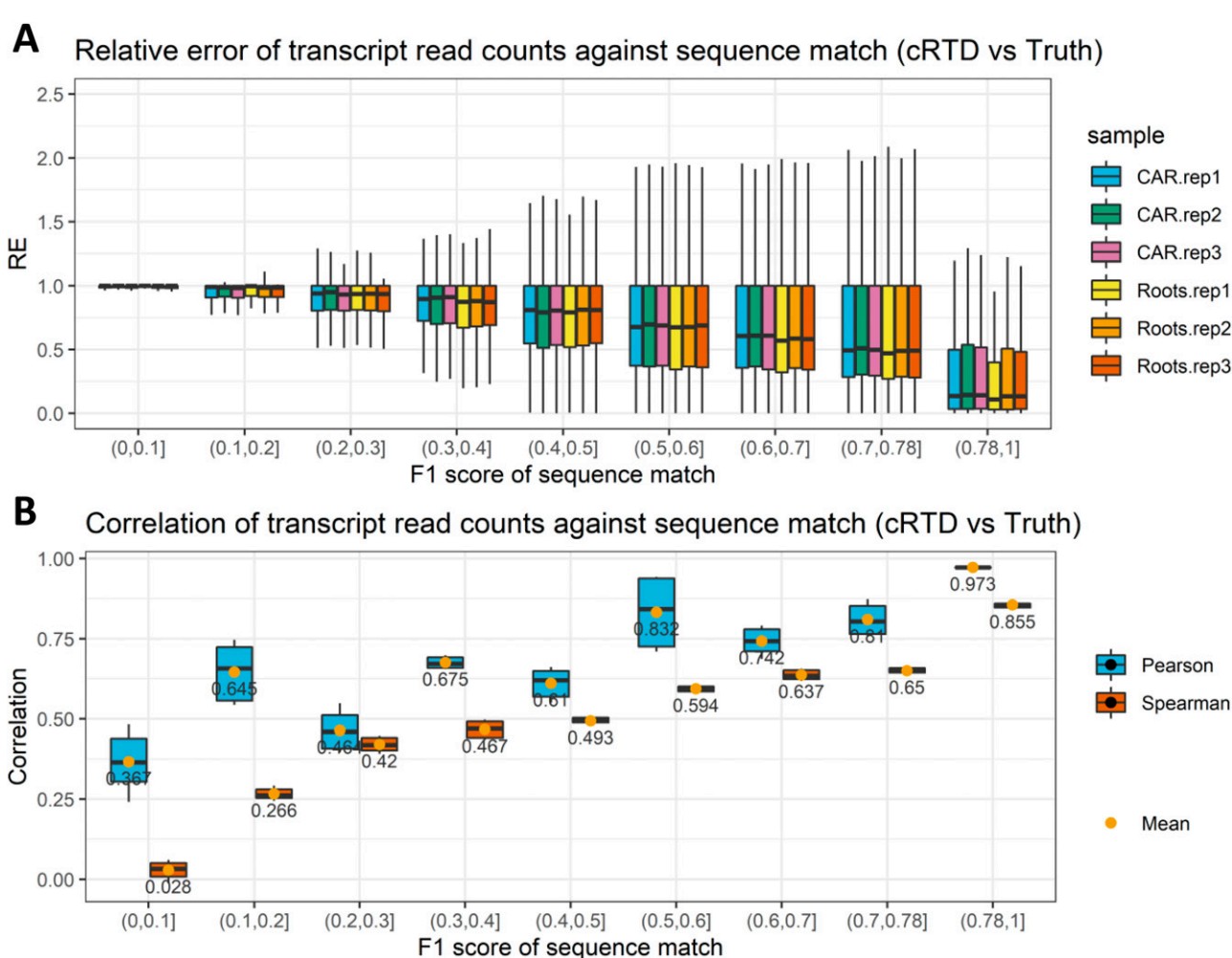

**Figure 6. Comparison of transcript quantification of cRTD and ground truth at different sequence similarities.**
F1 scores of sequence matches using Blastn are divided into different intervals. **(A, B)** The (A) relative errors and (B) Pearson and Spearman correlations of read counts are calculated for the matched transcripts within corresponding F1 score intervals. The outliers of the boxplots are removed.

between caryopsis and root tissues as well as the biological replicates of ground truth. The amount of the variation explained by PC1 and PC2 in all three datasets is not significantly different. Comparisons between DE genes, DAS genes, DE transcripts, and DTU transcripts illustrate that DE analysis at the gene level is most stable and least affected by the choice of RTD (Fig 7A). sRTD shows precision and recall both >99% for DE genes, whereas the cRTD only achieves 67.7% precision and 67.0% recall, missing about 4,000 DE genes whereas inferring 4,000 false-positive genes. The transcript level analysis is more variable, but the analyses for DAS genes and DTU transcripts both show the superior performance of sRTD over cRTD in both precision and recall, consistent with our observations that the use of the cRTD presents significant challenges for transcriptional level analysis.

We used the topGO R package to generate the enriched GO terms (*P*-value < 0.01) of genes and transcripts with significant expression changes in the categories of Biological Process, Molecular Function, and Cellular Component. The comparisons of significantly enriched GO terms indicate that sRTD has much higher recall and precision in

GO enrichment analysis (Fig 7B). It consistently predicts more GO information matched ground truth at both gene and transcript levels. For example, sRTD predicts 68, 27, 116, and 69 more GO terms from the DE genes, DAS genes, DE transcripts and DTU transcripts, which are not present in the cRTD results (Fig 7B). Many of the missing GO terms by cRTD are relevant to biological processes of responses and regulation, such as response to cold, insect, abscisic acid stimulus and stress, regulation of defense response and immune response, and various plasma membrane related processes (Table S10). Specifically, the GO terms regulation of response to salt stress (GO:1901000; DE genes), regulation of response to osmotic stress (GO:0047484; DE genes and DE transcripts), response to water (GO:0009415; DE transcripts), lateral root formation (GO: 0010311; DE genes and DE transcripts), and root cap development (GO:0048829; DE transcripts) associate with root formation and growth (Ogawa & Yamauchi, 2006; Habte et al, 2014; Alahmad et al, 2019; Berhin et al, 2019; Kreszies et al, 2020; Ouertani et al, 2021). The GO terms regulation of starch metabolic process (GO:2000904; DE transcripts), amylase activity (GO:0016160; DTU transcripts), pigment

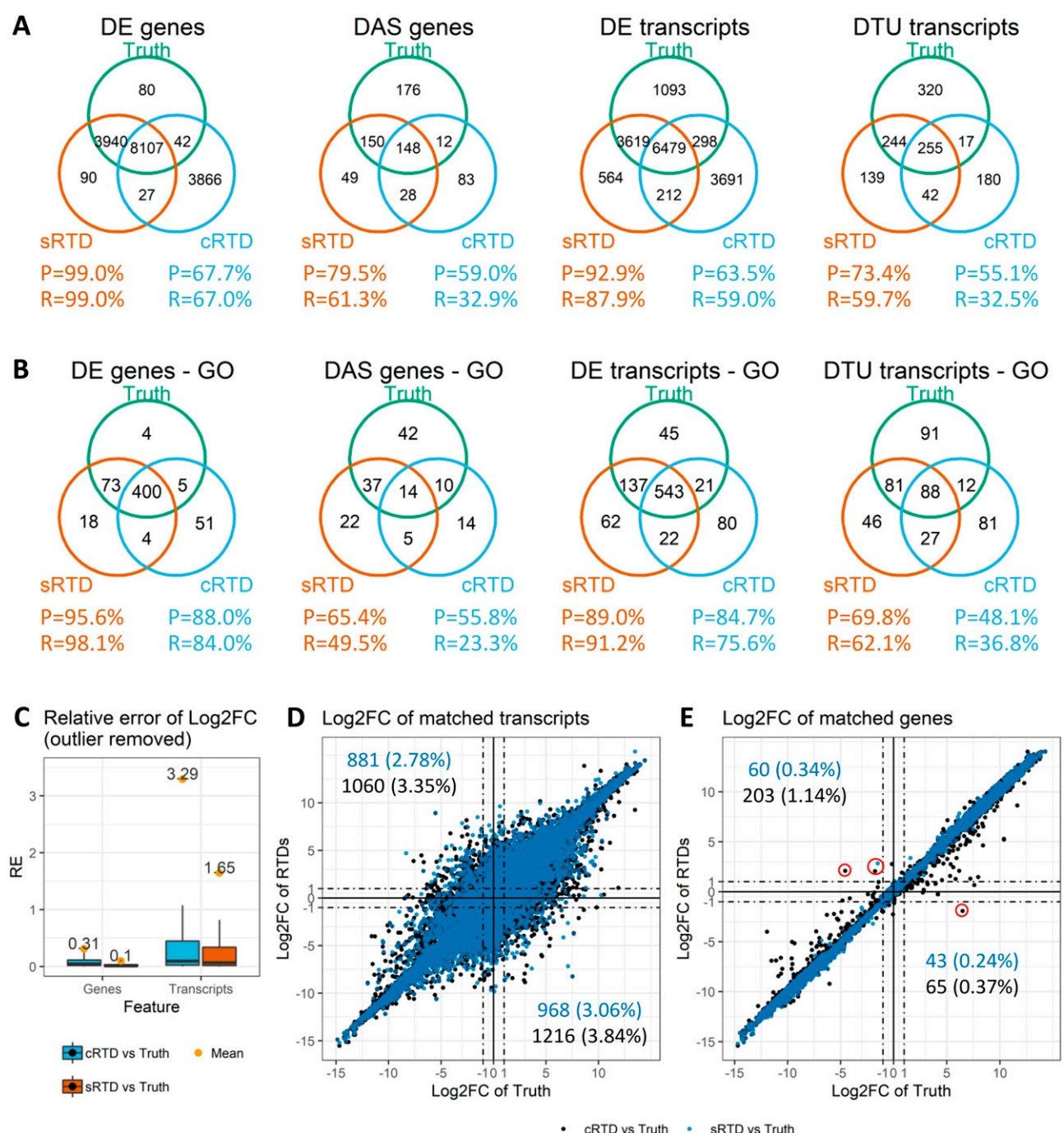

**Figure 7. Comparisons of differentially expressed and alternative spliced genes and transcripts.**
The transcripts between sRTD and cRTD were matched by sequence Blastn criteria F1 > 0.78 and Gffcompare with identical structure. The matched gene must have at least one matched transcript. The lowly expressed transcripts and genes in the ground truth, sRTD or cRTD were filtered, yielding 31,640 transcripts and 55,768 genes for comparisons. **(A)** Venn diagrams of DE genes, differentially alternatively spliced genes, DE transcripts and differential transcript usage transcripts in sRTD and cRTD compared with ground truth. R represents Recall and P represents Precision. **(B)** Venn diagrams of significantly enriched GO terms of DE genes, differentially alternatively spliced genes, DE transcripts, and differential transcript usage transcripts. **(C)** L2FC relative error distributions of matched genes and transcripts. **(D)** L2FC of matched transcripts. **(E)** L2FC of matched genes. In (D) and (E), the numbers and percentages on the top-left and bottom-right quadrants demonstrate the transcripts and genes that switch up- and down-regulations. The dashed lines highlight the L2FC cut-offs of 1 and −1 that we often use to determine the change significance. **(E)** The red circles in (E) highlight the genes that switch up- and down-regulation with |L2FC| ≥ 1.

metabolic process (GO:0042440; DAS genes) and pigment biosynthetic process (GO:0046148; DAS genes) are important to caryopsis development (Pagano et al, 1997; Lin et al, 2008; Shaik et al, 2016; Wang et al, 2020; Mackon et al, 2021). Therefore, the reductions of DE, DAS, and DTU genes and transcripts by using cRTD can also lead to a drop in important GO terms in the functional analysis.

We then compared the log$_2$-fold change (L2FC) differences of matched genes and transcripts using the quantification results obtained by using two RTDs and the ground truth. The L2FCs at the gene level are much closer to ground truth, with greater than one magnitude lower relative errors compared with the transcript level (Fig 7C). When comparing sRTD and cRTD to ground truth, we identified that 1,849 (5.84%) expressed transcripts in sRTD and 2,276 (7.19%) in cRTD have inverted up- and down-regulation (Fig 7D). At the gene level, the number of inversions significant decrease to 103 (0.58%) and 268 (1.51%) for sRTD and cRTD, respectively (Fig 7E). Applying a cut-off |L2FC| > 1, the numbers of regulatory switches are further reduced. For example, only two genes in cRTD and one gene in sRTD switch to up-regulation and one gene in cRTD to down-regulation.

## Discussion

Several studies also demonstrated that genotype-specific or individualised RTDs can improve some aspects of the transcriptomics analyses in different species other than barley. Petek et al (2020) constructed cultivar-specific RTDs in potato with a de novo assembly approach and used read mapping rate to cultivar-specific genome and transcriptome as the only evaluation metric to present the benefits of using cultivar-specific RTD. Munger et al (2014) incorporated known SNPs and INDELs into the common reference genome to create a pseudo strain-specific genome of the inbred mouse. The sequences variations were also integrated into inferred haplotype sequences of the founder mouse to construct an individualised diploid genome of outbred mice, which was a combination of two individualised haploid genome sequences. Then the offsets of the available transcriptome (a common reference RTD) were adjusted according to the two pseudo genomes to create individualised RTDs. The evaluation of simulated reads and real experimental data showed that read alignment, transcript quantifications and expression quantitative analyses were improved by using individualised RTDs in both inbred and outbred mice. However, this approach relies on modifying a common reference RTD with known sequence variations and has failed to address the comparisons of strain-specific AS events as well as dispensable genes and transcripts, which could account for, for example, up to 60% in crop species (Hirsch et al, 2014; Li et al, 2014; Golicz et al, 2016; Jin et al, 2016; Montenegro et al, 2017; Sun et al, 2017; Tao et al, 2019). In our study, we implemented more accurate evaluations by constructing the sRTD from the genotype-specific genome reference and RNA-seq data. We assessed the read mapping, transcript sequences and structures, AS events and expression quantitative analysis between sRTD and cRTD with the same parameters, providing more comprehensive insights into the superiority of genotype-specific genome reference for transcriptomics data analysis. By setting the reference genome as the

only variable and comparing the assembled sRTD to itself to estimate the technical variations, we were able to investigate the impacts of using a common reference on the outcomes of transcriptomic data analyses from different but related genotypes.

Although the assembled gene numbers were comparable in sRTD and cRTD, the use of a common reference genome resulted in a significant reduction in transcript diversity. STAR is a splice-aware aligner to map the RNA-seq reads to the reference genome and place the spliced reads over the introns, thereby can be used to determine the exon-intron boundaries and the AS sites (Dobin et al, 2013). The transcript isoforms and AS events are highly dependent on the correct identification of intron gap-open of RNA-Seq reads to the splice junctions of origin. We found that Barke RNA-seq read alignment to the Morex genome produced 11.6% fewer splice junctions which led to a >20% loss of introns and exons in the assembled cRTD. The reduction of splice junctions also resulted in >20% fewer AS events as well as >10% fewer transcript isoforms.

Predictably, the common transcripts of sRTD and cRTD have notably lower sequence variations (SNPs and INDELs) between the genomes of Barke and Morex (Fig 3). The sequence variations between different genotypes affect the transcriptome assembly on various levels. At the RNA-seq read mapping step, the sequence variations create barriers for the reads to find the correct genomic locations of origin. Higher sequence variations can lead to an increased number of unmapped reads in STAR (Dobin et al, 2013; Dobin & Gingeras, 2015). Our evaluation results (Table S3) as well as many studies in the literature show that map reads to a genotype-specific reference genome can greatly improve the read mapping rate and increase the RNA-seq data usage (Yuan & Qin, 2012; Munger et al, 2014; Petek et al, 2020). Most of the reference-based transcriptome assembly tools, such as Cufflinks (Trapnell et al, 2010), Scallop (Shao & Kingsford, 2017) and Stringtie (Pertea et al, 2015), generate splice graphs from the aligned reads with vertices representing exons and edges corresponding to splice junctions. The read coverage of exons and junctions are regarded as the weights to decompose the splice graph paths into optimal transcript models. The sequence variation can impact the identification of exon-intron boundaries as well as the number of aligned reads for weighting, thereby leading to changes of structure and diversity in the inferred transcript sets between sRTD and cRTD. Genotype variations, especially large deletions and insertions, also influence the read mapping to transcriptome for transcript quantification. It agrees with our findings that by using sRTD for quantification, the read mapping rate to transcript sequences is 2–3% higher than cRTD (Table S11).

sRTD improved the quantifications of transcript abundances and downstream differential expression analysis. We used simulated RNA-seq read data to compare transcript quantifications using cRTD and sRTD. We found that the quantifications of transcripts in sRTD have a higher correlation and lower errors to the ground truth. Not surprisingly, the quantification error increases and correlation decreases as the transcript sequence discrepancies increase between cRTD and sRTD (Fig 6). We also found that when the transcripts in cRTD have <40% sequence similarity to the transcripts in sRTD, their expression starts to be significantly lower (Fig S7A and B). The high expression often indicates better quality of the assembled transcripts. Therefore, a filter of low expressed transcripts

is an effective quality control to eliminate problematic transcripts. In addition, gene-level quantification of transcript abundance appears to be much more robust and less affected by the RTD with the relative error almost one magnitude lower compared with transcript level quantifications. As the designation of the common reference genome is arbitrary, we are also expecting to see improved quantifications for Morex RNA-seq data using Morex genome reference. In a recent study, we compared barley RTDs BaRTv2 (Barke-based assembly) and BaRTv1 (Morex-based assembly) using high-resolution RT-PCR data in Barke and Morex against the RTD quantifications using Barke and Morex RNA-seq data (Coulter et al, 2021 *Preprint*). With no exception, we see an increased accuracy when BaRTv1 was used on Morex RNA-seq data and when BaRTv2 was used on Barke RNA-seq data.

Quantification accuracy also directly affects the downstream differential expression analysis. Our comparisons indicated that with stringent filters of significance, only around 70% of the DE genes quantified using cRTD agree with ground truth. The transcript and AS level analyses have even lower precision (Fig 7A). We further investigated the expression of the overlapped and unique sets of genes and transcripts in the comparisons of Fig 7A. We found that the DE genes, DAS genes, DE transcripts, and DTU transcripts correctly predicted by using cRTD have higher expression (Fig S8). Our validation showed that the error of L2FC at the gene level is over one magnitude smaller than at the transcript level (Fig 7C). The error of L2FC is significantly higher and we have fewer correct identifications of DE transcripts, especially the AS related DAS genes and DTU transcripts (Fig 7A and D). At the gene level, there are only a few outlier genes that appear to switch between up- and down-regulation when assessed using the different RTDs (change sign of L2FC). The number gets further reduced when applying additional filters, for example, |L2FC| ≥ 1 (Fig 7E). The incorrect predictions of genes and transcripts with significant expression changes in cRTD also lead to the missing of important GO terms in the enrichment analysis, especially at AS level (Fig 7B and Table S10). Thus, in differential expression analyses, the high expressed transcripts and gene-level results can be ranked with higher confidence to accommodate a common reference.

Our evaluation indicated that after filtering the problematic transcripts, such as fragmented transcripts and transcripts with false splice junctions, we can have 70% correct assemblies by using a common reference. The precision is much higher than de novo approaches. In an evaluation study, Hsieh et al (2019) compared three state-of-the-art de novo assemblers Trinity (Grabherr et al, 2011), rnaSPAdescated (Bankevich et al, 2012), and Trans-ABySS (Robertson et al, 2010) with both simulated and real experimental data. The evaluation showed that these tools assembled up to 90% erroneous transcripts, including incomplete, over-extended, duplicated, and family-collapse (multiple transcripts were collapsed into a single contig in the assembly) transcripts. Freedman et al (2021) investigated the error, bias and noise of using Trinity (Grabherr et al, 2011) for de novo transcript assembly in mice. The assembled transcripts were highly fragmented, with an assembly error rate up to 83% at the nucleotide level. The quantification of assembled contigs failed to provide accurate analysis at the transcript level. Even at the gene level, the quantification had a fourfold or greater than the quantifications of using a benchmark

reference transcriptome, thereby, leading to increased false positives of downstream differential expression analysis. The reference-based approach in our study started with read alignment to a reference genome. It increases the efficiency and sensitivity and it can assemble low expressed transcripts (Martin & Wang, 2011; Conesa et al, 2016; Marchant et al, 2016). Thus, even if a genotype-specific reference is not available or in the studies of outbred genome, the assembly using a closely related reference genome is more adequate to provide accurate transcripts than de novo approaches.

With the development of cost-effective long read technologies, such as Oxford Nanopore and PacBio sequencing, and a repertoire of analysis tools, it is now more and more practical to generate a customised genome reference that could reflect the samples under investigation to make the best use of the data generated and achieved analysis with higher resolution and accuracy (Amarasinghe et al, 2020; Logsdon et al, 2020; Wang et al, 2021). In many cases, the common practice of using a single reference for one species is a method of convenience in the past due to various cost and technical reasons. Analysis using individualised genomes has gained great traction in human studies and is likely to cause a paradigm shift in future reference-based sequence analysis (Brittain et al, 2017; Rehm, 2017; Sherman & Salzberg, 2020). It has been reported that the throughput of both Nanopore and PacBio can reach 50–100 Gb per flowcell with a cost of <$50 per Gb (Logsdon et al, 2020). With this yield, a small number of flowcells is sufficient to sequence the full human genome. It is well worthwhile to add in the extra cost to construct the genotype-specific genome reference itself. Our study is one of the first studies to comprehensively examine the impact of genotype-specific genome reference on transcriptomics analysis. Our work presents the findings for the first step towards this new paradigm.

In summary, because of the difficulty in assembling transcripts accurately using short reads, transcript level analysis remains challenging. When using contemporary pseudo alignment methods for assessing transcript abundance, the impact of the RTD on quantification accuracy can be considerable for transcript-level analysis. However, reassuringly for most of the transcriptomic analyses, working at the gene level appears relatively robust and represents a reasonable, if potentially less informative, compromise when a high quality and comprehensive genotype-specific transcript reference is not available.

# Materials and Methods

### Data pre-processing and transcriptome assembly

Barke Illumina RNA-seq data from 20 tissues were taken from Coulter et al (2021) *Preprint*. The adapters on the raw RNA-seq reads were trimmed using Trimmomatic v0.39 (Bolger et al, 2014) with parameters chosen with the guidance of FastQC v0.11.9 (http://www.bioinformatics.babraham.ac.uk/projects/fastqc/). The trimmed reads in each sample were mapped to the Barke genome (Jayakodi et al, 2020) and Morex genome (Mascher et al, 2017) with STAR v2.7.3a (Dobin et al, 2013). We implemented the two-pass approach to increase the sensitivity of splice junction (SJ)

discovery. The minimum and maximum intron sizes were set to 60 and 15,000 in both passes. We allowed two mismatches at the first pass to improve the SJ detection sensitivity. The detected SJs in the first pass mapping were merged across samples and used as guidance for read alignment in the second pass. To reduce spurious alignments caused by sequence variation and compare the RTDs more precisely, mismatch was not allowed in the second pass of read mapping. We assembled the sRTD and cRTD in parallel using the same processes and parameters using Stringtie v2.1.4 (Pertea et al, 2015). The assemblies of 20 samples were merged with Stringtie-merge (Pertea et al, 2015). We applied various filters to achieve high-quality assemblies. (1) The transcripts with either non-canonical SJs or SJs with low read support (support criteria ≥3 uniquely aligned reads in ≥2 samples) were filtered out. (2) Redundant transcripts were defined as those with the same intron combinations but with different first and/or last exon lengths. Only the longest transcript amongst the same group of redundant transcripts was kept. (3) Transcript fragments <70% of the length of the longest in the same group of redundant transcripts were removed. To determine protein-coding transcripts, we extracted the transcript sequences of sRTD and cRTD from the Barke and Morex genomes with Gffread (Pertea & Pertea, 2020). We used Transuite, which identified authentic AUGs in transcript sequences, to translate them into protein sequences (Entizne et al, 2020 Preprint). We queried the protein sequences against the plant protein database UniProtKB for best-hit of annotated proteins using Blastp (Schneider et al, 2009). To identify AS, we used SUPPA2 to demonstrate the diversity of AS events of retained intron (RI), alternative 5′ splice site (A5), alternative 3′ splice site (A3), skipping exon (SE), alternative first exon (AF), alternative last exon (AL), and mutually exclusive exons (MX) (Trincado et al, 2018).

### Assessing the impact on transcript sequences and structures

We aligned the transcript sequences of both cRTD and sRTD (query) to transcript sequences of sRTD (target) with Blastn. Blasting sRTD against sRTD was used as a reference to estimate the frequency that Blastn fails to align a query sequence correctly. Each query returned only one hit with the best e-value and bit-score (e-value must <1.0 × $10^{-5}$ and Blastn parameter max_target_seqs = 1). If a target was aligned by multiple query transcripts, only the one/ones with the maximum matched bases were retrieved. To further evaluate the proportions of matches to the target and query lengths, we treated the query-to-target matched bases as true positives (TP) of sequence alignment. The unaligned bases of the target and query were false negatives (FN) and false positives (FP), respectively. The proportions of sequence overlap were evaluated with precision (TP/(TP+FP)) and recall (TP/(TP+FN)), and their weighted mean F1 score (2 × (Recall × Precision)/(Recall + Precision)) which took both recall and precision into account.

As sRTD and cRTD were assembled from read mappings to different reference genomes, their coordinates and structures cannot be compared readily. To compare the structures of transcripts at equivalent genome coordinates, we aligned the transcript sequences of both cRTD and sRTD to the Barke genome with Minimap2 v2.17 (Li, 2018), denoting the results as cRTD.minimap2 and sRTD.minimap2. Mis-aligned transcripts were filtered if the

fragments of a transcript aligned to multiple chromosomes or strands and the aligned transcripts did not meet the criteria of minimum intron size 60 and maximum intron size 15,000. Transcript structures were evaluated by querying the genome coordinates of cRTD.minimap2 and sRTD.minimap2 against target sRTD at levels of nucleotide bases, introns, intron chains (combination of introns), exons, transcripts and gene loci by using GffCompare v0.11.2 (Pertea & Pertea, 2020). SUPPA2 was also used to investigate AS events and percent spliced-ins (PSIs) in annotations of cRTD.minimap2 and sRTD.minimap2 (Trincado et al, 2018). The AS events and PSIs of these two annotations were compared with that of sRTD. At the AS event level, we defined True Positives (TP) as the shared events between query (cRTD.minimap2 or sRTD.minimap2) and target (sRTD), False Positives (FP) as the events only in the query and False Negatives (FN) as the events only in the target. Then, precision and recall were used to measure the agreement between query and target. The AS event PSI values were calculated based on the transcript per million reads (TPMs) from the Salmon quantification of sRTD.minimap2 and cRTD.minimap2 with Barke RNA-seq reads from all 20 samples. In these analyses, the comparison of sRTD.minimap2 to sRTD provided a reference to estimate technical errors in Minimap2, GffCompare, and SUPPA2.

### Assessing the impact on the accuracy of transcript quantification

To quantify the absolute quantification accuracy using both RTDs, we also used simulated RNA-seq reads to compare the quantifications between sRTD and cRTD at both transcript and gene levels. Specifically, the numbers of simulated reads for the sRTD transcripts were directly specified according to a read count matrix. To have replications and make the simulation results reflect real experimental data as much as possible, we used the read count matrix from the transcript quantification of "caryopsis" and "root" tissues, which was a subset of the Barke RNA-seq seven-tissue data in Jayakodi et al (2020), each with three biological replicates. We used the Polyester R package to simulate RNA-seq data for caryopsis and roots tissues each with three replicates, 150-bp paired-end reads, 40–60 million reads per replicate from the sRTD transcript sequences according to the read count matrix (Frazee et al, 2015). Then we quantified the cRTD and sRTD with Salmon by using the simulated RNA-seq reads (Patro et al, 2017). The gene and transcript quantifications of two RTDs were compared, in terms of Pearson and Spearman correlation and relative errors, to the ground truth read count matrix used to generate the data.

### Assessing the impact on differential gene and AS analysis

We applied the 3D RNA-seq pipeline to analyse the DE genes and transcripts, differential alternative splicing (DAS) and differential transcript usage (DTU) based on three datasets (Guo et al, 2020). These were sRTD and cRTD quantifications from the simulated RNA-seq reads and the read count matrix, which is the ground truth for the simulation. In all analyses, we set the contrast group for testing expression changes to "caryopsis versus root." The significance was determined with a Benjamini-Hochberg (BH) adjusted P-value < 0.01 and absolute $\log_2$-fold change ≥1 for DE genes and transcripts and BH adjusted P-value < 0.01 and absolute Δ precentage spliced

($\Delta PS$) ≥ 10% for the DAS genes and DTU transcripts. Gene Ontology (GO) annotations of genes and transcripts in sRTD and cRTD were generated by PANNZER (Törönen & Holm, 2021). GO enrichment analyses of DE genes, DAS genes, DE transcripts and DTU transcripts were performed by using the topGO R package with significance criteria *P*-value < 0.01 (Alexa et al 2006). To compare the quantification at the individual gene and transcript level, we used the results of transcript sequence and structure comparisons to match the equivalent individuals.

# Data Availability

The RNA-seq data used in this study are available at SRA with BioProject accession number PRJNA755156. The scripts of the data analysis can be viewed from: https://github.com/wyguo/genotype_specific_RTD.

# Supplementary Information

# Acknowledgements

This work was jointly supported by funding from the Biotechnology and Biological Sciences Research Council (BBSRC) BB/R014582/1 to R Waugh and R Zhang; BB/S020160/1 to R Zhang and R Waugh; BB/S004610/1 (16 ERA-CAPS BARN) to R Waugh; the Scottish Government Rural and Environment Science and Analytical Services division (RESAS) to W Guo, R Zhang, and R Waugh.

## Author Contributions

W Guo: conceptualization, data curation, formal analysis, investigation, visualization, methodology, and writing—original draft, review, and editing.
M Coulter: data curation and writing—original draft.
R Waugh: conceptualization, data curation, funding acquisition, and writing—original draft, review, and editing.
R Zhang: conceptualization, supervision, funding acquisition, methodology, project administration, and writing—original draft, review, and editing.

## Conflict of Interest Statement

The authors declare that they have no conflict of interest.

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
