## [Reviewer comments · Life Science Alliance]

The value of genotype-specific reference for transcriptome analyses in barley

Wenbin Guo, Max Coulter, Robbie Waugh, and Runxuan Zhang

DOI: 10.26508/lsa.202101255

Corresponding author(s): Dr. Runxuan Zhang (The James Hutton Institute)

Review timeline:

Submission Date:	2021-10-05
Editorial Decision:	2021-12-01
Revision Received:	2022-02-24
Editorial Decision:	2022-03-25
Revision Received:	2022-04-10
Accepted:	2020-04-11

Scientific Editor: Novella Guidi

Transaction Report:

No Peer Review Process File is available with this article, as the authors have chosen not to make the review process public in this case.

Re: Life Science Alliance manuscript #LSA-2021-01255-T

Runxuan Zhang
The James Hutton Institute

Dear Dr. Zhang,

Thank you for submitting your manuscript entitled "The value of genotype-specific reference for transcriptome analyses" to Life Science Alliance. The manuscript was assessed by expert reviewers, whose comments are appended to this letter. As you will note from the reviewers' comments below, all the reviewers are quite enthusiastic about the work but they do raise important concerns that would need to be addressed in the revised version before resubmission with the common one being the lack of proper analyses to fit with the proposed broad scope. We, thus, encourage you to submit a revised version of the manuscript back to LSA that responds to all of the reviewers' points including conducting comprehensive evaluation of how much the availability of cRTD or the lack of sRTD in an RNA-seq analysis would affect the outcome. The authors should use multiple assembly strategies in addition to the one shown in Figure 1 to see whether the results observed in this study are biased, as suggested by Reviewer 1. Please also address Reviewer 2 concerns by conducting the proposed additional analyses and Reviewer 3 concerns on whether specific categories of genes /transcripts are enriched among those showing the main differences between the two assemblies both in terms of gene novelty and transcript quantification.

Thank you for this interesting contribution to Life Science Alliance. We are looking forward to receiving your revised manuscript.

Sincerely,

- A letter addressing the reviewers' comments point by point.
- An editable version of the final text (.DOC or .DOCX) is needed for copyediting (no PDFs).
- High-resolution figure, supplementary figure and video files uploaded as individual files: See our detailed guidelines for preparing your production-ready images, <https://www.life-science-alliance.org/authors>
- Summary blurb (enter in submission system): A short text summarizing in a single sentence the study (max. 200 characters including spaces). This text is used in conjunction with the titles of papers, hence should be informative and complementary to the title and running title. It should describe the context and significance of the findings for a general readership; it should be written in the present tense and refer to the work in the third person. Author names should not be mentioned.
- By submitting a revision, you attest that you are aware of our payment policies found here: <https://www.life-science-alliance.org/copyright-license-fee>

B. MANUSCRIPT ORGANIZATION AND FORMATTING:

March 25, 2022

RE: Life Science Alliance Manuscript #LSA-2021-01255-TR

Dr. Runxuan Zhang
The James Hutton Institute
Information and Computational Science
Errol Road, Invergowrie
James Hutton Institute
Dundee, Angus dd2 5da
United Kingdom

Dear Dr. Zhang,

Thank you for submitting your revised manuscript entitled "The value of genotype-specific reference for transcriptome analyses". We would be happy to publish your paper in Life Science Alliance pending final revisions necessary to meet our formatting guidelines.

- we encourage you to revise the figure legends for figures 1 and S7 such that the figure panels are introduced in alphabetical order;
- reference in the supplementary file should be part of the main references
- Please upload all figure files as individual ones, including the supplementary figure files; all figure legends should only appear in the main manuscript file
- please add the Twitter handle of your host institute/organization as well as your own or/and one of the authors in our system
- please note that the titles in the system and manuscript file must match
- please add callouts for Figures 3A-B; S6A-C and S7A-B to your main manuscript text

To upload the final version of your manuscript, please log in to your account:
<https://lsa.msubmit.net/cgi-bin/main.plex>

A. FINAL FILES:

B. MANUSCRIPT ORGANIZATION AND FORMATTING:

Sincerely,

RE: Life Science Alliance Manuscript #LSA-2021-01255-TRR

Dr. Runxuan Zhang
The James Hutton Institute
Information and Computational Science
Errol Road, Invergowrie
James Hutton Institute
Dundee, Angus dd2 5da
United Kingdom

Dear Dr. Zhang,

Thank you for submitting your Research Article entitled "The value of genotype-specific reference for transcriptome analyses in barley". It is a pleasure to let you know that your manuscript is now accepted for publication in Life Science Alliance. Congratulations on this interesting work.

DISTRIBUTION OF MATERIALS:

Again, congratulations on a very nice paper. I hope you found the review process to be constructive and are pleased with how the manuscript was handled editorially. We look forward to future exciting submissions from your lab.

Sincerely,
